# Digital PCR characterizes epithelial cell populations in murine duodenal organoids

**Karla Acosta-Virgen**[1], **Hugo David González-Conchillos**[1], **Gabriela Vallejo-Flores**[2], **Lizbeth Iliana Salazar-Villatoro**[1], **Ernesto Guerrero-Sánchez**[3], **Adolfo Martínez-Palomo**[1] and **Martha Espinosa-Cantellano**[1]*

1 Departamento de Infectómica y Patogénesis Molecular, Center for Research and Advanced Studies, Mexico City, Mexico, 2 Department of Molecular Biology, Max Planck Institute for Infection Biology, Berlin, Germany, 3 QIAGEN MEXICO, Mexico City, Mexico

* mespinosac@cinvestav.mx

## Abstract

Three-dimensional cultures are powerful tools to recapitulate animal and human tissues. Under the influence of specific growth factors, adult stem cells differentiate and organize into 3D cultures named organoids. The molecular phenotyping of these structures is an essential step for validating an organoid model. However, the limited number of organoids generated in culture yields very low amounts of genetic material, making phenotyping difficult. Recently, digital PCR (dPCR) techniques have become available for the highly sensitive detection of genetic material at low concentrations. The aim of this work was to apply dPCR to the identification of the various cell populations expected to be present in murine duodenal organoids. Results show the potential use of dPCR as a genetic characterization tool for organoids.

## Introduction

Organoids are three-dimensional cultures that have proven very useful in biomedical research due to the preservation of structure and cell composition of the tissues of origin. Using intestinal adult stem cells isolated directly from living tissues, which retain the ability to reproduce and differentiate into tissue-specific structures, it is possible to replicate the tissue microenvironment in the laboratory. The biological relevance of three-dimensional models has greatly impacted biomedical research. In particular, organoid models can recapitulate the *in vivo* architecture, functionality, and genetic signatures from original tissues [1–4].

Gastrointestinal tract organoids (esophagus, stomach, small intestine, and colon) have been developed; all of them are driven by a rapid turnover rate and strong differentiation capabilities [5]. Adult intestinal stem cells (ISC) can generate crypt-villus units, with self-renew capabilities that allow differentiation into all types of intestinal cells under specific *in vitro* conditions [6–10]. All cell populations from the epithelia of origin should be represented in organoids; in the case of the small intestine, they should include stem cells, amplifying cells, absorptive enterocytes, goblet cells, enteroendocrine cells, and Paneth cells. Differentiation is determined by the exposure to different growth factors added to the culture medium, which are a representation of the intrinsic factors that are produced in the intestinal niche [6,11–13].

**Data availability statement:** All analyzed data generated in this study are included in this article and in Supporting information files.

**Funding:** KAV was supported with a doctorate scholarship No. 481481 from the Consejo Nacional de Humanidades, Ciencias y Tecnologías (CONAHCYT). This work was supported by CONAHCYT grant No. CF 2019/2558586 Frontera de la Ciencia to MEC. The funders had no role in study design, data collection and analysis, decision to publish, or preparation of the manuscript.

**Competing interests:** The authors have declared that no competing interests exist.

Establishing an organoid model is a complex process that involves manipulating every variable in the culture conditions until long-term cultures are established, and finally characterizing the organoid structurally, genetically, and functionally to validate the resemblance to the tissue of origin, all of this, in minute Matrigel drops. Once the model is established, characterization of the organoid has been mainly addressed using bright-field imaging, immunofluorescent staining, and transmission and scanning electron microscopy.

Identification of tissue-specific cell populations has been frequently carried out through quantification by RT-PCR and qPCR of representative genes. However, these analyses require the use of calibration curves before performing each experiment, increasing the cost and introducing bias in the quantification of epithelial markers for the characterization of the model [14–19]. The low amounts of residual RNA in Matrigel, likely derived from extracellular vesicles found within the extracellular matrix (ECM), and the low concentration of RNA in organoids have prompted researchers to search for more efficient RNA isolation protocols or even pool several Matrigel drops to identify poorly expressed epithelial markers in their samples [20].

Digital PCR (dPCR) is a very sensitive technique capable of detecting low-abundance targets among high concentrations of other nucleic acids in the sample, due to thousands of partitions of the reaction mixture randomly distributed in the nanowells. Collecting the amplification data of each nanowell allows quantification to proceed individually and, by applying Poisson statistics, it is possible to calculate the number of copies per microliter present in the sample [21,22]. Each nanowell contains a fraction of the PCR mix, which will be analyzed and classified as a positive partition if it at least contains one copy of the DNA sequence since it will have a higher fluorescence than those partitions that do not have a copy of nucleic acids, which are classified as negative partitions. dPCR shows advantages over qPCR in terms of simplicity, reproducibility, lower limit of detection, and absolute quantification [21,22]. For those reasons, we present here the use of dPCR for the identification of epithelial markers in murine duodenal organoids as an alternative for the molecular characterization of three-dimensional models in samples with low concentrations of genetic material, combined with classical techniques of characterization, such as confocal- and transmission electron microscopy.

## Materials and Methods

### Isolation and culture of duodenal crypts

The animal research protocol was submitted to the Internal Committee for the Care and Use of Laboratory Animals (CICUAL-Cinvestav) and approved under number 0318-21. Sex indistinct 6-8 weeks-old ICR-(CD-1) mice were provided by the institutional animal facility (UPEAL-Cinvestav, under approved protocol number 0318-21). Cervical dislocation as euthanasic methods were applied, and the small intestine was removed to obtain 2-3 cm of the duodenum. A portion of the duodenum was preserved in TRIzol™, paraformaldehyde (PFA) or glutaraldehyde and used as tissue of origin control. The rest of tissues were washed five times with sterile PBS (Phosphate Buffered Saline) supplemented with antibiotics (penicillin-streptomycin 1%, 50 μg/ml gentamycin, and 1.25 μg/ml amphotericin B), cut into very tiny sections, and incubated in 1X chelating buffer (prepared in Milli-Q water, added with 0.5M DTT, 0.5M EDTA pH 8, and sterile filtered; S1 Supporting information) for 45 min at 37°C under shaking. Sections were placed in a cell strainer (70 μm), squeezed to obtain crypts, and cells collected in a tube with basal medium: Advanced DMEM/F12 (Dulbecco's Modified Eagle Medium), added 2 mM Glutamax and 10 mM HEPES; supplemented with 5% of Fetal Bovine Serum (FBS: Biowest, BIO-S1620-500). The cell solution was centrifuged

(750 g/5 min/4°C) and the supernatant was discarded. The pellet containing isolated duodenum crypts was resuspended in thawed Matrigel (Corning, 356231) and seeded in 40μl drops on a pre-warmed 24-well plate. After 15 min, the polymerized Matrigel drops were covered with 500 μl of organoid culture medium (see below), which was changed every 3 days. Cells were kept in a humidified incubator (5% $CO_2$ and 37°C) for six days until passage day.

## Organoid culture and maintenance

To grow intestinal organoids, duodenal crypts containing adult Intestinal Stem Cells (ISC) from ICR-(CD-1) mice were isolated using previously reported protocols [7,9,23,24]. $5X10^5$ cells were seeded into each Matrigel drop and grown in organoid culture medium, that was prepared on basal medium (Advanced DMEM/F12 (Gibco, 12634028) supplemented with 2mM Glutamax (Gibco, 35050-061), 10 mM 4-(2-hydroxyethyl)-1-piperazineethanesulfonic acid (HEPES; Gibco, 15630080), 1% penicillin/streptomycin (Biowest, L0022-100)) supplemented with 1X B27 and 1X N2 supplements (Gibco, 17504-044 and 17502-048 respectively). Complete organoid culture medium requires basal medium, added with 50% conditioned Wnt3A-medium and 10% conditioned R-spondin1 medium, and supplemented with 20 ng/ml murine epidermal growth factor (Invitrogen, PMG8043), 150 ng/ml murine noggin (Peprotech, 250-38), 150 ng/mL human fibroblast growth factor (FGF)-10 (Peprotech, 100-26), 1.25 mM N-acetyl-L-cystein, 10mM nicotinamide, 1 mM TGF-β RI Kinase Inhibitor IV (Calbiochem, A83–01). Rhoc inhibitor Y-27632 (Stemcell, 72304) was added only the first 3 days of culture.

Duodenal organoids were passaged every six days, until they increased their size five-six times. For passage, Matrigel drops were recovered from the plate using cold basal medium, transferred into Falcon tubes, and centrifuged (750 g/5 min/4°C). The supernatant was discarded, 400μl TrypLE Express enzyme (Gibco, 12604013) per drop was added to the pellet and incubated for 5 min at 37°C. Basal medium supplemented with 5% FBS was added to inactivate the enzymatic activity. Organoids were carefully sheared using an insulin needle syringe, transferred into a Falcon tube, added with 5ml basal medium supplemented with 5% FBS; cells were counted, and centrifuged (750 g/5 min/4°C). The pellet was resuspended in Matrigel, seeded onto 24-well plates ($5 \times 10^4$ cells/40 μl Matrigel drops), covered in organoid cultured medium, and kept in a humidified incubator (5% $CO_2$ and 37°C). Medium was changed every 3 days.

## RNA isolation and cDNA synthesis

Organoids were classified as early (Eo) after 1-3 passages, intermediate (Io) after 4-6 passages, and late (Lo) after 7 passages or more. Duodenal tissue (≈0.3 cm) or one Matrigel drop containing Eo, Io, or Lo organoids were placed in 500 μL of TRIzol™ reagent (Invitrogen, 15596026), vortexed vigorously, and kept at -80°C until processing. Total RNA (50μl) was extracted using RNeasy Mini Kit (QIAGEN, 74104) following the manufacturer's recommendations. RNA quantification was performed by fluorometric analysis using the Qubit RNA HS Assay Kit (Invitrogen, Q32852) according to the manufacturer's protocols. RNA concentrations oscillated between 7.9 and 103.30 ng/μl. cDNA was synthesized using SuperScript™ III First-Strand Synthesis SuperMix (Invitrogen, 11752050) and quantified using the Qubit 1X dsDNA HS kit (Invitrogen, Q33230). cDNA concentrations oscillated between 12.2 and 40 ng/μl.

Specific primers were designed for epithelial markers using the software Primer3Plus (https://primer3plus.com/cgi-bin/dev/primer3plus.cgi) with default configuration for qPCR experiments. Primer sequences and NCBI gene accession numbers are listed in S2 Table.

## Digital PCR

Reaction mixtures were prepared using the QIAcuity EG PCR Kit (QIAGEN, 250111) for organoids at different growth stages and tissue samples, and dPCR was performed on a QIAcuity Digital PCR System (QIAGEN, Hilden, Germany). Sample mixes were placed on QIAcuity Nanoplate 8.5k 24-well plates (QIAGEN, 250011); each well had a final volume of 12 µl of reaction mix containing 4µl of cDNA. Thermocycling was performed under the following conditions: polymerase activation at 95 °C for 2 min, amplification for 35 cycles of 95 °C for 15 s, 59 °C for 15 s and 72° for 60 s. Finally, the cycling step was 40° for 5 min and the plate was kept at 4° once running was ended. The imaging step was set to the green channel with 500 ms of exposure and gain 6. The readouts of positive and negative partitions were analyzed via the QIAcuity Software Suite. Each partition was classified as either negative or positive for any marker, based on a manual threshold amplitude, expressed Relative Fluorescent Units (RFU). Raw data was exported and scatter plots for each set of samples were graphed using Rstudio and the ggplot2 package. The QIAcuity One manual was used to apply the threshold for every sample analyzed.

## Transmission Electron Microscopy

Slices of duodenum tissue and duodenal organoids were fixed in 2.5% glutaraldehyde and post-fixed with 1% osmium tetroxide in 5mM cacodylate buffer 0.1 M for 1h. Samples were dehydrated with increasing concentrations of ethanol (50, 70, 90 and 100%), followed by propylene oxide treatment. Preinclusion was performed using propylene oxide in combination with epoxy resin (2:1, 1:1 and 1:2) and inclusion on pure resin (Polysciences, 08792) was carried out at 60° overnight. Ultrathin sections (60-90 nm) were obtained and stained with uranyl acetate, lead citrate and Ruthenium red. Finally, sections were mounted in 300 mesh copper TEM grids and observed using a JEOL JEM 1011- transmission electron microscope (JEOL Ltd. Tokyo, Japan).

## Immunofluorescence and Confocal Microscopy

For immunostainings, sections of duodenal tissue (≈0.3 cm) or one Matrigel drop containing organoids were fixed in 1% paraformaldehyde (1h) and embedded in paraffin according to the conventional histological technique. Five mm-thick sections were mounted on silanized-coated slides. Samples were dehydrated and dewaxed with pre-heated antigen-retrieval solution (Tris-EDTA buffer: 10 mM Tris base, 1 mM EDTA solution, 0.05% Tween 20, pH 9.0) for 30 min. Samples were incubated with blocking solution (5% donkey serum + 1% FBS in PBS buffer) for 1h, washed, and incubated overnight at 4°C with primary antibodies (dil 1:100 in blocking buffer); after washing, samples were incubated with secondary antibodies for 1h (dil 1:400 in PBS-Tween 20), and nuclei were stained with Hoechst (Thermo Scientific, 62249) (1:1000 in PBS-Tween 20). Stained samples were mounted in Mowiol (Sigma-Aldrich, 81381) and observed in an LSM900 confocal microscope (Carl Zeiss, Germany). Details of the antibodies used are included in S3 Table.

## Results

To monitor the presence of the different cell populations over time, growing organoids were classified into three stages: early organoids (Eo), from passages 1 to 3; intermediate organoids (Io), from passages 4 to 6, and late organoids (Lo), from passages 7 to forward. Epithelial gene markers were tested in each stage by dPCR, and ultrastructure was analyzed by transmission electron microscopy. Several markers were corroborated by confocal microscopy.

## Identification and quantification of duodenal cell markers in organoids at different stages through cDNA amplification by dPCR

In order to evaluate the expression of the different biomarkers, RNAs extracted from early, intermediate, and late duodenal organoids were translated into their corresponding cDNAs and tested by nanowell-based digital PCR for the identification of the following cell population markers: LGR5 and SOX9 (stem cells), PCNA (proliferating cells), Mucine-2 (goblet cells), DCKL1 (Tuft cells), ChA (enteroendocrine cells), LYS (Paneth cells), EPCAM (epithelial cells), Villin-1 (mature enterocytes). GAPDH was used as endogenous control. Analysis was performed in three independent replicates (M1: mouse 1, M2: mouse 2, and M3: mouse 3).

Even when the assay was designed to evaluate the qualitative identification of the marker, one of the main features of dPCR is its ability to provide quantification of target sequences without the need for reference material. Our results reveal that expression of the different cell markers is heterogeneous and independent of culture passage, as shown by the copies/μl measured for each marker (Fig 1). Stem cells were identified through the expression of LGR5 and SOX9. LGR5 was expressed at very low levels in most samples tested, in particular in early passages, where detection as low as 0.385 and 1.518 copies/μl was possible. In contrast, SOX9 was expressed at 11.26 to 2081.4 copies/μl in the same cells. Proliferating cell nuclear antigen (PCNA) was used as a proliferating cell marker; higher values were recorded in early passages, reaching up to 5656.5 copies/μl, while intermediate and late passages recorded 409.1 and 423.4 copies/μl, respectively. Goblet cells were evidenced by the expression of Mucin 2. A variable range of expression was observed: for M1, the highest (2566.3 copies/μl) and lowest value (2.128 copies/μl); nevertheless, the expression was also lower compared to other markers.

Doublecortin-like kinase 1 (Dclk1) expression, used to identify intestinal Tuft cells, did not reach 100 copies/ μl in any sample, the highest quantity was observed in the late passage of M2 and the lowest in the intermediate passage of M3 with 1.42 copies/μl. Chromogranin A (ChA) was used as a marker for mature enteroendocrine cells. The lowest value was identified at Io in M1 (7.106 copies/μl) and at Eo in M1 (1.391 copies/μl), and the highest value was in M3, with 632 copies/μl. Paneth cell marker lysozyme was tested; in Eo, 175 copies/μl were identified for M1, being the highest value for this marker in all samples; on the other hand, the lowest values were 4.18 copies/μl for the Lo in M1 and 2.243 copies/μl for the M3, both in the Lo passages.

Epithelial cells were followed analyzing the expression of epithelial cell adhesion molecule (EpCAM), which had the highest number of copies in all samples and all passages. In early organoids, the values ranged between 3612.4 to 10446.3 copies/μl, for the intermediate organoids between 1036.3-1635.4 copies/μl, and in late organoids, between 551.2-1160.6 copies/μl, being 551.2 copies/μl the lowest quantity for this marker. The expression of the enterocyte marker gene, Villin-1, in early organoids ranged from 377.9 to 755.9 copies/μl, registering higher numbers in Io for M1 and M2, with more the 800 copies/μl, but not in M3, with 75.68 copies/μl. In late organoids, the values ranged from 22.63 to 150 copies/μl, being the lowest values for enterocyte detection. As expected, the highest values were registered in the endogenous control, GAPDH, with the lowest value of 3348.5 copies/μl found in Eo in M1, and the highest value of 23309.6 copies/μl for the Lo in the M2. Absolute quantification is graphed in Fig 1 and absolute values are shown in S1 Table.

## Positive partitions in dPCR shows heterogenicity in the expression of cell population markers in organoids at different passages

Besides registering the number of copies per microliter, the distribution of positive partitions counting the expression of markers in all nanowells in the three independent experiments

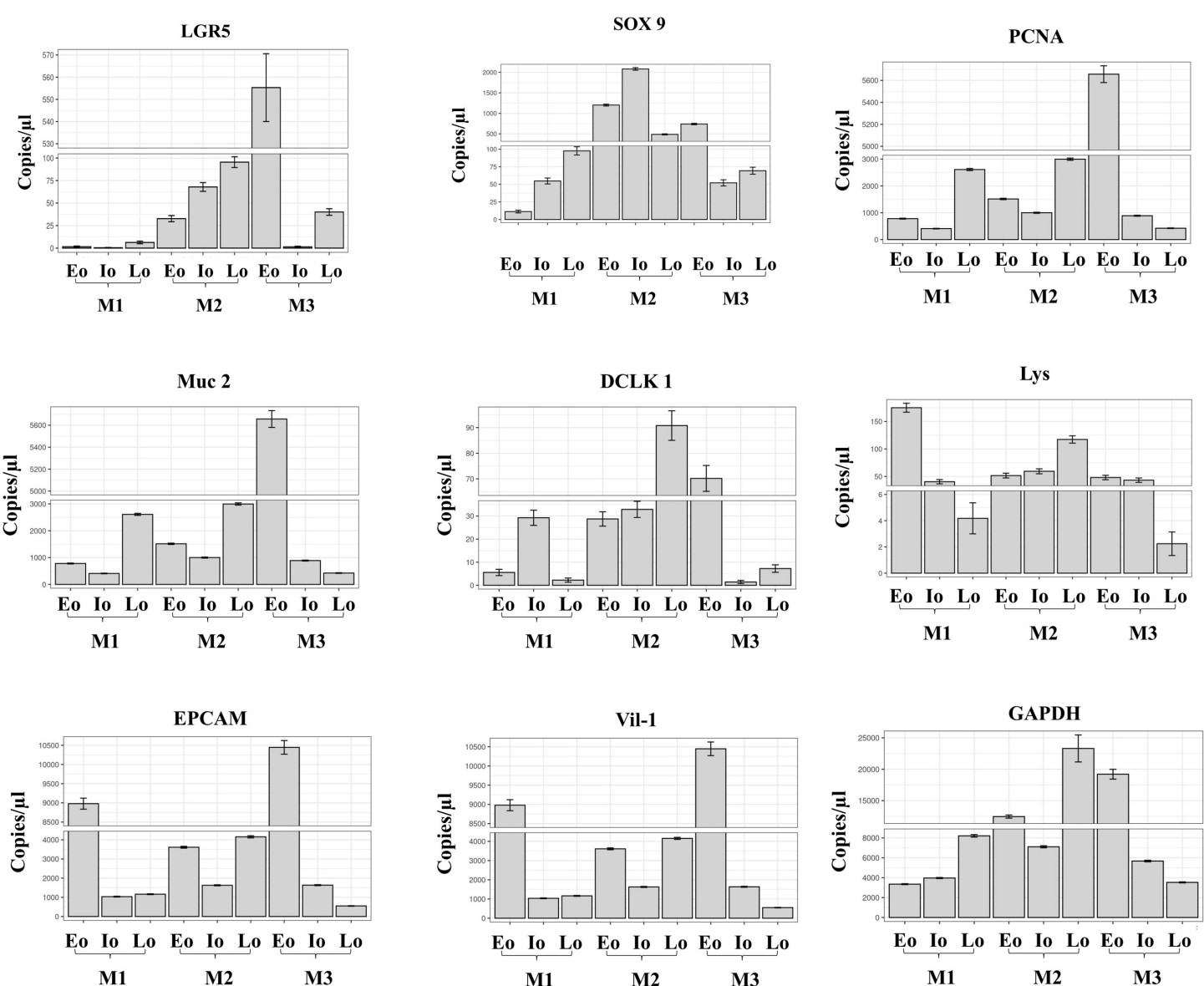

**Fig 1. Identification of duodenal cell markers in organoids and their quantification by dPCR.** The heterogeneity of the expression of epithelial markers was observed independently of the culture passage stage. Expression is quantified in copies/µl. Abbreviations: Eo: early organoids; Io: intermediate organoids; Lo: late organoids; M1: mouse 1; M2: mouse 2; M3: mouse 3.

was performed. After amplification in nanowell dPCR, each reaction (partition) is analyzed to determine whether it contains the target sequence, since amplification renders fluorescence above the threshold. By counting the number of positive and negative amplifications (below the threshold), the concentration of the target sequence in the original sample can be accurately determined. The fluorescence intensity (RFU) was measured in all positive partitions.

The best representation of positive partitions that are comparable to the tissue of origin, was observed in the amplification of genes PCNA, EPCAM, and the endogenous control GAPDH, for all passages in the three independent experiments. LGR5 expression is low in all samples, except for early organoids in M3. In the case of SOX 9, the highest expression was evident in M1 tissue and Eo of M2. More positive partitions were detected in the tissues

for MUC2, as well as in Eo from M2 and Io in M3. Also, Vill-1 was more evident in tissues; however, a high rate of positive partitions was observed in all early organoids and in the intermediate organoids from M2. On the other hand, Lys was also well detected in all tissues and expressed consistently in all organoid samples from M2. ChA-positive partitions were observed in tissue and early organoids from M3. In contrast, very few positive partitions were identified for the DCLK-1 marker in all samples, including the tissues. In Fig 2 we show representative scatterplots of all markers obtained for LGR5, PCNA and EPCAM from M2. Individual scatterplots of all samples and markers obtained from mice 1 to 3 are presented in S1-S10 Figs.

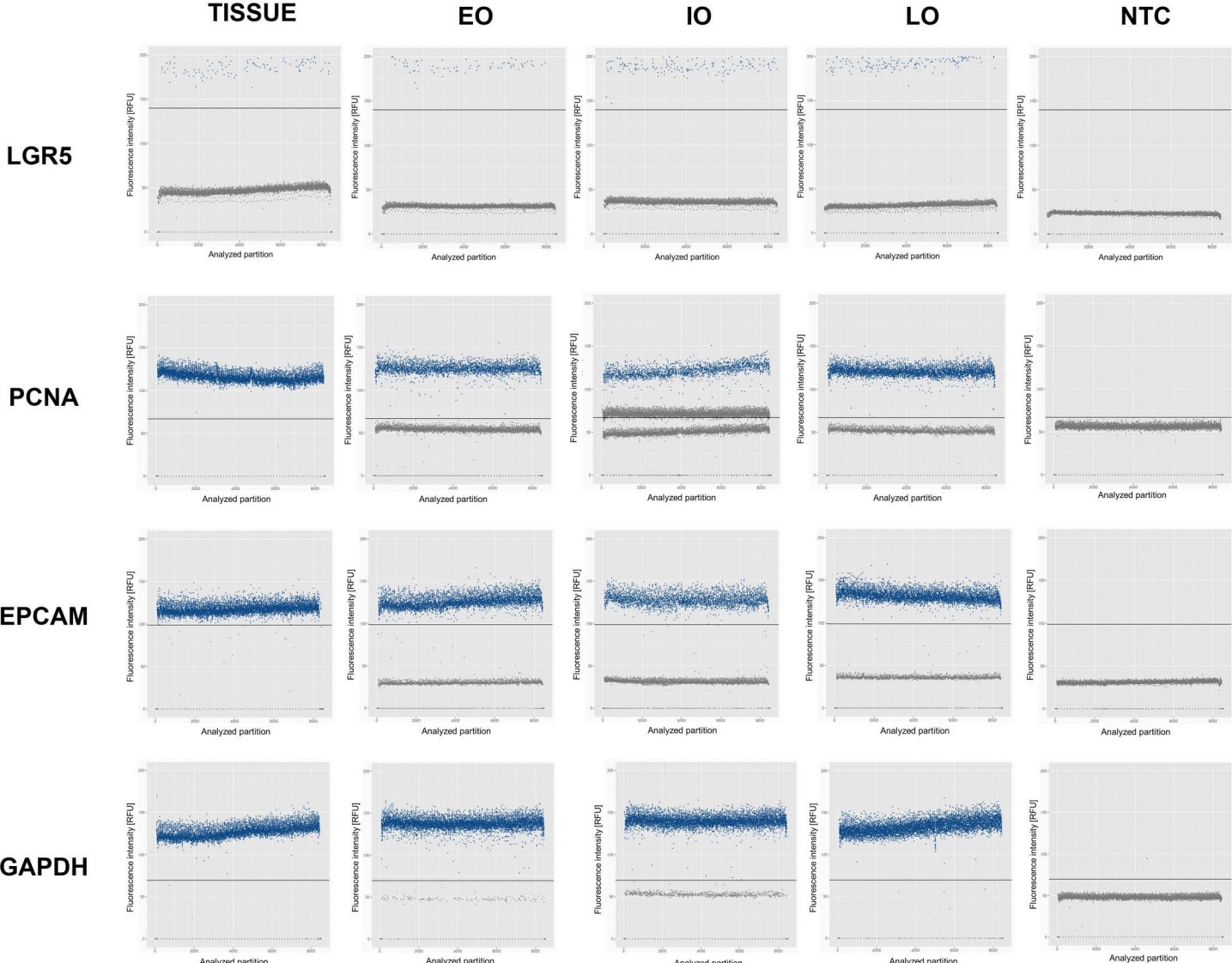

**Fig 2. Representative scatterplots of dPCR of epithelial markers in tissue and duodenal organoids (LGR5, PCNA, EPCAM and GAPDH).** Amplification of the target gene is registered in each nanowell through fluorescence intensity: positive amplification is recorded above the threshold, while negative amplification falls below the threshold. Each dot represents the fluorescence intensity recorded in one nanowell. Positive partitions for all epithelial markers were graphed for all growth times in three independent experiments. Abbreviations: Eo: Early organoids; Io: Intermediate organoids; Lo: Late organoids; NTC: No template control.

## Ultrastructural characterization of murine duodenal organoids

Comparative transmission electron microscopy between duodenal murine tissue and organoids at different culture passages, was performed (Fig 3). Duodenum extracted from mice presents the conventional architecture described for the mature tissue: enterocytes, goblet cells, epithelial cells, and prominent microvilli were observed (Fig 3A). Organoids at different passages (Eo, Io and Lo) showed shape irregularities, but well-organized cells were observed.

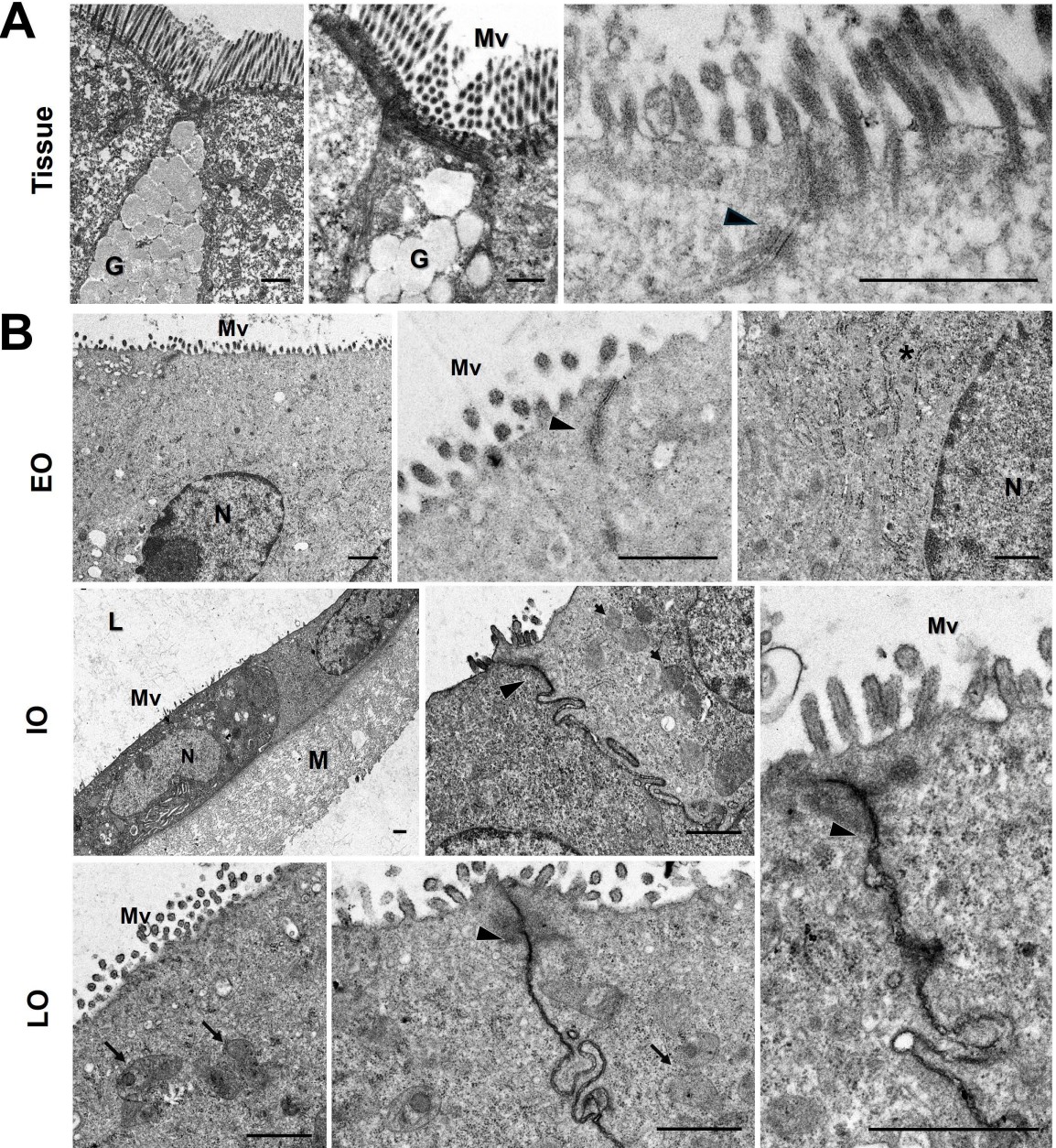

**Fig 3. Ultrastructure of murine duodenal organoids.** (A) Representative images of duodenal tissue, and (B) murine duodenal organoids observed by transmission electron microscopy. Integrity of the epithelium was observed in tissue and organoids. Abbreviations: G: Goblet cells; L: Lumen; M: Matrigel; Mv: Microvilli; N: Nucleus; Tj: Tight junctions. *: Ribosomes; Arrows: mitochondria; Arrowheads: tight junctions. Scale bar: 1 μm.

Mature enterocytes with microvilli, evident tight junctions, and well-defined internal structuration (Fig 3B), were comparable to those observed in the tissue of origin.

## Validation of epithelial markers in duodenal organoids

To confirm the identity of duodenal cell populations within the organoids, we selected functional epithelial markers and compared their presence at three culture stages with the tissue of origin. Lgr5 and Sox 9 markers were used to identify stem cells. Microvilli were identified using an anti-villin1 antibody in combination with E-cadherin, a main component of adherent junctions; this combination was identified in both original tissue and organoids (Fig 4). Also,

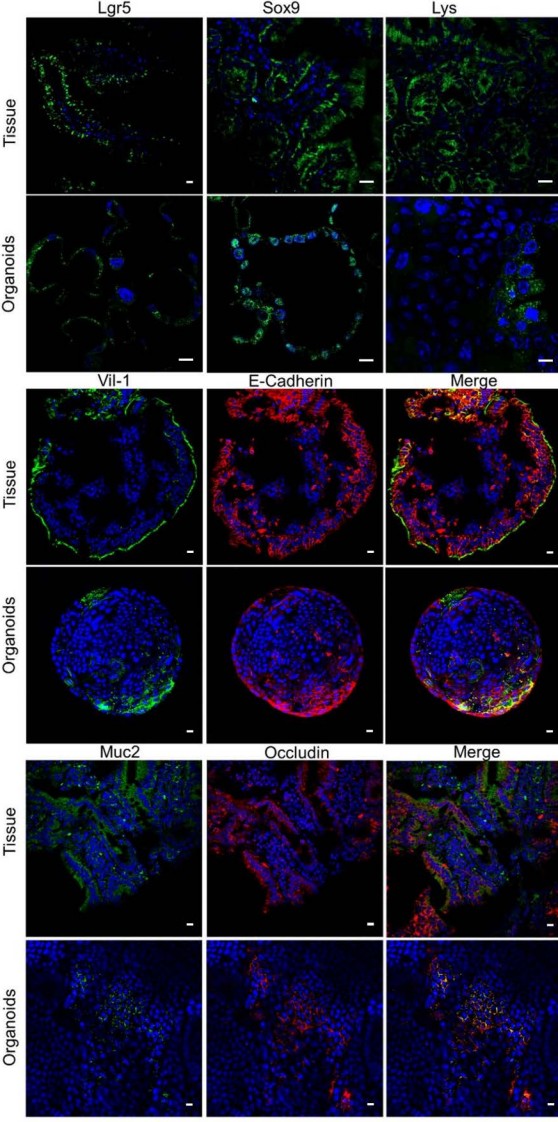

**Fig 4. Duodenal organoids express epithelial markers of the main cell populations in tissues.** A) Immunostaining of stem cells (Lgr5 and Sox 9), microvilli (Vil-1), Paneth cells (Lys), goblet cells (Muc2) and tight junctions (E-cadherin and occludin) in tissue and murine duodenal organoids labeling specific markers of these cell types. Scale bar: 10 μm.

goblet cells were identified using an anti-Muc2 antibody in combination with an anti-occludin marker, a major component of tight junctions. Our results confirm the presence of basic epithelial populations in duodenal organoids, as identified by dPCR.

## Discussion

Since their development [14,23,24], Haga clic o pulse aquí para escribir texto.intestinal organoids have been used as models to study various parasitic, bacterial, and viral infections. During the Covid-19 pandemic, the study of SARS-CoV2 infection in three-dimensional models of several human organs proved a powerful tool to analyze the physiopathology of the infection and allowed key insights to understand the symptomatology and progression of the disease [25–28].

Organoids derived from adult stem cells (ASC) resemble the characteristics of the tissue from which they originate; epithelial markers are key to evidence the presence of native cell populations in those three-dimensional models. Small intestine cell markers for native populations are well-defined in organoids. Stem cells populations are mainly identified using Lgr5, Olfm4 + , CD24 + or Sox9 + markers [14,16,17,29–33]. Proliferative cells express Proliferating Cell Nuclear Antigen (PCNA) as a marker [15]. EPCAM expression is restricted to epithelia in adult tissues. Villin-1 is an epithelial marker for microvillar actin filaments, localized in the apical border of the small intestine [34,35].

Using murine duodenum, we developed duodenal organoids grown during different passages and characterize this model. First, we established three stages of growth; we refer to Early organoids (Eo) to those grown between 1 to 3 passages; Intermediate organoids (Io) to those between passages 4-6 and Late organoids (Lo) from passages 7 forward.

Structurally, our organoids present characteristics similar to tissue of origin. Thin sections of organoids at early, intermediate, and late passages, show features of mature enterocytes, epithelial cells, and prominent microvilli, demonstrated by transmission electron microscopy (Fig 3). Duodenal organoids expressed occludin, a typical epithelial marker of tight junctions, E-cadherin, present in adherent junctions, and β-catenin, a protein involved in the cell-cell adhesion (Fig 4). Those markers are important for the intestinal barrier and selective mucosal permeability in vertebrate tissues [36–38]. We also demonstrate the presence of tight junctions in murine duodenal organoids by TEM.

Once our organoids were grown, we performed specific cDNA identification and evaluated the expression through nanowell-based digital PCR (dPCR, QIAcuity from Qiagen) to identify various cell populations in our cultures. In dPCR, targets are randomly distributed in all available partitions. Poisson statistics distribution will estimate the average number of molecules per partition and measure the number of copies per positive partition. Positive partitions will display a fluorescent signal, over the total number of partitions, and this information is used to calculate the absolute concentration of the target gene in the sample [39].

Organoid cultures are highly variable and heterogeneous due to multiple factors, such as different experimental conditions or extraction methods directly reflected in the purity and conservation of the genetic material that is isolated from a single drop of Matrigel. Reports in the literature of the extraction of mRNA from organoids to analyze gene expression do not describe concentrations of mRNA in a single drop of Matrigel or the number of drops of Matrigel-embedded organoids were pooled for the characterization of epithelial markers. Our organoids contained minute amounts of mRNA. We decided to use dPCR to counteract the low concentration of genetic material. Surprisingly, we were able to detect eight different markers in duodenal murine organoids at different passages of growth.

Taking one Matrigel drop, we extracted RNA and synthetized cDNA from organoids in different passages (Eo, Io, Lo), obtained from duodenal crypts containing adult stem cells in three independent experiments. In S1 Table) we show that even at very low cDNA concentration, dPCR is capable of detecting and quantifying the cDNA in each reaction, for example, for LGR5 gene, at early passages, 0.385 copies/μl for M1 were detected, and up to 95.45 copies/μl in M2, suggesting that even when the LGR5 gene is expressed at a very low ratio during early passages, it is sufficient to induce proliferation of stem cells, which can be detected at higher concentrations in organoids undergoing further passages.

An example of the variability in expression in intestinal organoid cultures is the case of the stem cell markers: LGR5 and SOX9. It has been reported that the expression of these markers can vary depending on different experimental conditions, such as the composition of the culture medium, the presence of signaling factors from conditioned media, the mouse strain from which the intestinal crypts were extracted, the extraction methods, the number of cell passages, the days of culture and even the ability to handle the samples. Such variability can influence the functional properties of the developed organoids, and their ability to model certain diseases or respond to treatments.

Digital PCR has revolutionized the identification of genetic material using the high sensitivity to detect DNA, even in samples with low concentrations of genetic material [40,41].

Both digital PCR (dPCR) and droplet digital PCR (ddPCR) are methods for absolute quantification of nucleic acids without the need for standard curves; however, both use distinct approaches. Regarding partitioning techniques, dPCR could be considered more versatile, due to partitioning in each PCR reaction being divided into many individual reactions, which are amplified using microfluidic chips or nanowell plates, as is used in the QIAcuity technology, the dPCR technology of QIAGEN. On the other hand, ddPCR needs droplets to partition for the PCR reaction to occur; a single reaction mixture is emulsified into thousands or millions of droplets, each containing a small portion of the sample. Regarding genetic material, in dPCR, the quantification is per each partition, which contains at least one copy of the target DNA or RNA material when the partition is positive, and the absolute quantification is based on the number of positive partitions after amplification; in contrast, in ddPCR, each droplet acts as an individual PCR reaction, hence here the absolute quantifications rely on the number of positive droplets detected after amplification [42,43].

Many applications have been developed for dPCR and ddPCR. However, until now, ddPCR has been more frequently used than dPCR. For example, ddPCR has been useful for the detection of gene mutations with tumorigenic potential in stem cell-based therapeutics. Park and collaborators consider that ddPCR has higher sensitivity and accuracy in detecting gene mutations in human induced pluripotent stem cell-derived cardiomyocytes and recommend this genetic detection tool over conventional qPCR [44]. In infectious diseases, ddPCR has shown a great potential when it is fully integrated on a lab-on-a-disc (LOAD) system. Zhang et al., developed a device for screening representative clinical viruses (Influenza A, respiratory syncytial, varicella zoster, Zika and adenovirus) for absolute quantification of molecular genetic products with diagnostic purposes [45]. Screening of drug targets for human diseases is another field where digital PCR is used. *Ptgfr*, a cardiac G-protein-coupled receptor, was identified as a novel drug target using ddPCR in a rat model [46].

Recently, ddPCR has been used to determine copy number (CN) heterogeneity in metastatic site tumors and in pooled, single, and invading organoids, using clinically relevant target genes such as FGFR1, ADGRA2, NSD3, and PAK1. The authors show intra-and inter-organoid genetic heterogeneity and highlight the potential of this method to functionally validate the role of FGFR1 in metastatic development, which has remarkable applications in clinical studies [47].

On the other hand, nanowell dPCR has been used to quantify intact proviruses, showing sensitivity to detect down to 1-10 copies of HIV DNA [48]. Likewise, dPCR testing for fecal contamination of agricultural products have shown its potential application in the field, by successfully detecting *Bacteroides* markers from dog, cow, human and cat pools with a surprisingly low detection limit of 250 fg/µL in strawberries inoculated with fecal material [49]. In addition, dPCR has been used to detect gene expression, mutations, copy number variation, pathogens, residual contamination, and quality monitoring in public health, to mention a few [43,47–51].

In this study, we highlight some advantages of dPCR technology: 1) *higher precision and sensibility*, identifying even a few copies of genetic material in samples with low concentrations of DNA or RNA, in comparison with conventional PCR techniques, 2) *absolute quantification* in dPCR offers the possibility to know the number of copies per microliter of each sample, allowing precise comparisons between different samples and independent experiments, 3) *Reduce technical variability*, because sample manipulation is minimum and calibration curves are not necessary, 4) Analysis of heterogeneous samples, as they contain several cell populations, organoids are usually difficult to analyze individually; however, with dPCR, it is possible to identify different epithelial markers from minimal sample quantities. We here propose dPCR as a useful tool for the characterization of tissue-specific cell markers in organoids.

## Supporting Information

**S1 Supporting information.  Preparation of chelating buffer.**
(DOCX)

**S1 Table.  Quantification of target sequences of duodenal cell populations.**
(DOCX)

**S2 Table.  Primer sequences and NCBI gene accession numbers.**
(DOCX)

**S3 Table.  Antibodies used for confocal microscopy detection of specific antigens.**
(DOCX)

**S1 Fig.  Scatterplots of all samples showing the distribution of positive partitions of LGR5 gene amplifications for the identification of stem cells.**
(TIF)

**S2 Fig.  Scatterplots of all samples showing the distribution of positive partitions of SOX 9 gene amplifications for the identification of stem cells.**
(TIF)

**S3 Fig.  Scatterplots of all samples showing the distribution of positive partitions of PCNA gene amplifications for the identification of proliferating cells.**
(TIF)

**S4 Fig.  Scatterplots of all samples showing the distribution of positive partitions of MUC2 gene amplifications for the identification of Goblet cells.**
(TIF)

**S5 Fig.  Scatterplots of all samples showing the distribution of positive partitions of DCLK1 gene amplifications for the identification of Tuft cells.**
(TIF)

**S6 Fig. Scatterplots of all samples showing the distribution of positive partitions of CHGA gene amplifications for the identification of enteroendocrine cells.**
(TIF)

**S7 Fig. Scatterplots of all samples showing the distribution of positive partitions of LYS gene amplifications for the identification of Paneth cells.**
(TIF)

**S8 Fig. Scatterplots of all samples showing the distribution of positive partitions of EPCAM gene amplifications for the identification of epithelial cells.**
(TIF)

**S9 Fig. Scatterplots of all samples showing the distribution of positive partitions of VIL gene amplifications for the identification of microvilli.**
(TIF)

**S10 Fig. Scatterplots of all samples showing the distribution of positive partitions of GAPDH gene amplifications as an internal positive control.**
(TIF)

## Acknowledgments

We would like to acknowledge Gabriela Hernández-Pizano, Leonardo Carrasco-Mote, Karla Monserrat Gil-Becerril, Brenda Vargas-Hernández and Daniel Talamás-Lara for their excellent technical assistance. This work was conducted in accordance with the animal facility guidelines of the Comité Interno para el Cuidado y Uso de los Animales de Laboratorio (CICUAL-Cinvestav), under approved protocol number 0318-21.

## Author contributions

**Conceptualization:** Karla Acosta-Virgen, Hugo David González-Conchillos, Gabriela Vallejo-Flores, Adolfo Martínez-Palomo.

**Data curation:** Karla Acosta-Virgen, Ernesto Guerrero-Sánchez.

**Formal analysis:** Karla Acosta-Virgen, Ernesto Guerrero-Sánchez.

**Investigation:** Karla Acosta-Virgen, Lizbeth iliana Salazar-Villatoro, Martha Espinosa-Cantellano.

**Methodology:** Karla Acosta-Virgen, Hugo David González-Conchillos, Lizbeth iliana Salazar-Villatoro.

**Project administration:** Martha Espinosa-Cantellano.

**Resources:** Martha Espinosa-Cantellano.

**Supervision:** Adolfo Martínez-Palomo, Martha Espinosa-Cantellano.

**Writing – original draft:** Karla Acosta-Virgen, Martha Espinosa-Cantellano.

**Writing – review & editing:** Karla Acosta-Virgen, Hugo David González-Conchillos, Gabriela Vallejo-Flores, Lizbeth iliana Salazar-Villatoro, Ernesto Guerrero-Sánchez, Adolfo Martínez-Palomo, Martha Espinosa-Cantellano.

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
