## [Decision Letter · Decision Letter 0]

14 Oct 2024

PONE-D-24-39881Digital PCR technology identifies epithelial markers in murine duodenal organoidsPLOS ONE

Dear Dr. Espinosa-Cantellano,

Thank you for submitting your manuscript to PLOS ONE. After careful consideration, we feel that it has merit but does not fully meet PLOS ONE’s publication criteria as it currently stands. Therefore, we invite you to submit a revised version of the manuscript that addresses the points raised during the review process.

We look forward to receiving your revised manuscript.

Kind regards,

Alvaro Galli

Academic Editor

PLOS ONE

Journal Requirements:

 “KAV was supported with a doctorate scholarship No. 481481 from the Consejo Nacional de Humanidades, Ciencias y Tecnologías (CONAHCYT). This work was supported by CONAHCYT grant No. CF 2019/2558586 Frontera de la Ciencia to MEC.”

6. Please include a copy of Table 1 which you refer to in your text on page 17.

7. Please include captions for your Supporting Information files at the end of your manuscript, and update any in-text citations to match accordingly. Please see our Supporting Information guidelines for more information: http://journals.plos.org/plosone/s/supporting-information .

Reviewers' comments:

Reviewer's Responses to Questions

**Comments to the Author**

1. Is the manuscript technically sound, and do the data support the conclusions?

Reviewer #1: Partly

Reviewer #2: Yes

Reviewer #3: Partly

2. Has the statistical analysis been performed appropriately and rigorously? 

Reviewer #1: No

Reviewer #2: N/A

Reviewer #3: I Don't Know

3. Have the authors made all data underlying the findings in their manuscript fully available?

Reviewer #1: Yes

Reviewer #2: Yes

Reviewer #3: Yes

4. Is the manuscript presented in an intelligible fashion and written in standard English?

Reviewer #1: Yes

Reviewer #2: Yes

Reviewer #3: Yes

5. Review Comments to the Author

Reviewer #1: The aim of the manuscript is to identify key cellular components of murine duodenal tissue in organoid preparations using digital PCR (dPCR) technique. The results include dPCR results as well as electron microscopic and confocal characterization of duodenal organoids collected at different stages and accordingly classified into early, intermediate and late organoids. The results are presented with clarity and are understandable on scientific and language levels. The methods used in the manuscript are described in details and allow reproduction of the results obtained. Nevertheless, some concerns need to be addressed by the authors and are discussed in the following.

Major comments

1. The dPCR technique has been recently used on mouse organoids (Lake et al., 2024; 10.3389/fcell.2024.1358583). Although this does not necessarily affect the novelty of the approach of the authors to the study of duodenal organoids, the paper should be cited and eventual differences should be discussed. Also, the authors should consider discussing the novelty associated to the use of microscopy techniques for the characterization of duodenal organoids (Fig.3 and Fig.4). Is the characterization of early, intermediate and late duodenal organoids new? If yes, from which point of view?

2. The authors compare the dPCR technique with RT-PCR and qPCR but do not perform any of the latter to compare their results. We think the manuscript would benefit from at least an exemplificative comparison between the two methods to support the novelty and preference of the authors´ approach.

3. Regarding the dPCR results, no specific statistical analysis is provided. The authors discuss variability across mice but it is not clear if the variability derives from biological differences or if it is intrinsic to the method used. To compare this, technical replicates are required on a per-sample basis. If the material is not sufficient, one should at least compare organoids from the same animal and not different animals or time-points. Without this, the comparisons discussed in the paper across animals and organoid stage are not statistically solid. Also, GADPH was detected as housekeeping gene but not used for any normalization in the results.

4. Several points can be addressed on the figures: (a) Fig.1 was probably loaded incorrectly since it is the same as Fig.3. Therefore no comment can be added on Fig. 1, which nevertheless seems to be the main finding of the whole manuscript; (b) Fig.2 shows different epithelial markers as the one included in the figure legend. Also, it does not include any “No template control” as introduced in the figure legend. (c) In Fig.3 no scale bar is present and it is not discussed what the different arrows should point to. (d) Scale bars are also missing in Fig.4. (e) The authors should consider discussing Fig.3 and Fig.4 in more details in order to justify their appearance as main figures and therefore as main result of the manuscript. In the present state of the manuscript, no quantification is provided of the represented markers and the novelty of the results is not completely clear. If the figures only represent a confirmation of the expression of the markers, then it may be more fitting to include them as supplementary figures.

Minor points

1. In the Supplementary material 2 not all information about the used antibodies are listed.

Reviewer #2: 1. Title:

It could be beneficial to slightly reword the title for clarity and engagement, such as:

- "Identification of Epithelial Markers in Murine Duodenal Organoids Using Digital PCR Technology."

- "Digital PCR Characterizes Epithelial Cell Populations in Murine Duodenal Organoids.

2. Materials and Methods

- A few areas could benefit from more specific information, such as the concentrations of growth factors used in the culture medium, which are crucial for reproducibility

- Several abbreviations are used without first providing the full terms. For instance, abbreviations such as DMEM (Dulbecco's Modified Eagle Medium), PBS (Phosphate Buffered Saline), and others should be introduced with their full names upon first mention. I recommend revising the text to include the full names of all abbreviations when they are first introduced.

- Antibody incubation steps are clear, but the concentrations or dilutions of both primary and secondary antibodies should be included to ensure reproducibility.

- The concentration of Hoechst stain used in the experiments should be specified.

Add Specific Concentrations/Volumes: In several places, concentrations (especially for growth factors and antibodies) are missing. Including these values would make the protocol more precise.

3. Supplementary material 2

- The size of the PCR product should be specified in base pairs (bp) for clarity and precision.

- The oligonucleotide sequences of the primers must be indicated in the 5′ to 3′ orientation to ensure correct interpretation and reproducibility.

Reviewer #3: There are several issues that need revision. 1) In the methods part, it is not clear at all how the authors proceeded to grow organoids once they had isolated and cultured duodenal crypts. It is not enough to refer to other publications without making clear what procedure the authors used - at least this was unclear for me. Please make this unambigously clear in the manuscript.

2. I definitely recommend to show the results obtained for GAPDH in the manuscript itself and NOT ONLY in the supplemental material. The extent of GAPDH expression is a crucial control and definitely needs to be shown so that readers can immediately compare the results for GAPDH with those obtained for cell specificically expressed genes.

3. Unclear is, to what extent the organoids contained regular cells and to what extent they showed irregularities. Here the authors have to show what percentage of cells was regular and what percentage irregular, or how many structures (percentage-wise) were seen compared to original tissue. Also, here I would recommend to state whether these percentage values are in the range of the results that other authors have observed or not. A reader who is not that familiar with how organoids should look like would definitely need this information.

4. In the last part of the manuscript (results/discussion), the authors name ddPCR, dPCR and QIAcuity dPCR. They do not explain what the main difference is. Though, it is crucial - otherwise the readers do not understand why the authors are claiming that their data and results are significant and worth to be published and made know to the scientific community. The differences between ddPCR, dPCR, QIAcuity dPCR need to be clarified with a few sentences so that also readers who do not use these methods can clearly understand the main differences.

5. The legends needs to be more informative. What is shown exactly? in the micrographs it should be stated in the legend to what structures the arrows point to and why that is important.

6. PLOS authors have the option to publish the peer review history of their article (what does this mean? ). If published, this will include your full peer review and any attached files.

**Do you want your identity to be public for this peer review?** For information about this choice, including consent withdrawal, please see our Privacy Policy .

Reviewer #1: **Yes: ** Davide Gobbo

Reviewer #2: **Yes: ** Huda Hisham Sultan Alkatib

Reviewer #3: No

---

## [Author Response · Author response to Decision Letter 0]

29 Nov 2024

Reviewer #1:

The aim of the manuscript is to identify key cellular components of murine duodenal tissue in organoid preparations using digital PCR (dPCR) technique. The results include dPCR results as well as electron microscopic and confocal characterization of duodenal organoids collected at different stages and accordingly classified into early, intermediate and late organoids. The results are presented with clarity and are understandable on scientific and language levels. The methods used in the manuscript are described in details and allow reproduction of the results obtained. Nevertheless, some concerns need to be addressed by the authors and are discussed in the following.

Major comments

1. The dPCR technique has been recently used on mouse organoids (Lake et al., 2024; 10.3389/fcell.2024.1358583). Although this does not necessarily affect the novelty of the approach of the authors to the study of duodenal organoids, the paper should be cited and eventual differences should be discussed. Also, the authors should consider discussing the novelty associated to the use of microscopy techniques for the characterization of duodenal organoids (Fig.3 and Fig.4). Is the characterization of early, intermediate and late duodenal organoids new? If yes, from which point of view?

Thank you for the suggestion. We discuss the work of Lake et al., 2024 in lines 439-444 and include the reference in line 604. This group shows very interesting results using ddPCR that could have important clinical applications. However, our work is more focused on basic research using dPCR. We further extend the differences in the use of ddPCR and dPCR in our response to Reviewer No. 2 in lines 410 to 424.

Organoids have proven important tools to emulate the tissues of origin in the lab. To confirm the similarity to the tissues of origin, they are subject to many different analyses, including microscopical techniques. In our work, electron microscopy allows us to identify specific ultrastructural characteristics of the duodenal epithelium, such as the tight junctions that are crucial to maintain the barrier integrity, or the microvilli, which are specialized structures of this tissue (Fig. 3). Through confocal microscopy we were able to locate biomarkers of the different cell populations characteristic of the small intestine (Fig. 4). With these results we were confident that our organoids mimic the intestinal tissue.

We thank you for highlighting the distinction of early, intermediate, and late duodenal organoids. This was implemented to monitor the expression of biomarkers over time, which would allow us to assess whether or not there were changes in cell populations in the different passages. Text has been added in line 206 to reflect this novelty.

2. The authors compare the dPCR technique with RT-PCR and qPCR but do not perform any of the latter to compare their results. We think the manuscript would benefit from at least an exemplificative comparison between the two methods to support the novelty and preference of the authors´ approach.

In the beginning of this project, we attempted to amplify 5 genes (EPCAM, SOX9, PCNA and VILLIN, normalized to GAPDH expression) by qPCR in our growing organoids. Because of the low concentrations of isolated RNA, we were unable to amplify target genes consistently and, in some cases, no amplification was observed. One problem is that qPCR requires the preparation of several replicas, which consume large amounts of Matrigel drops. The results obtained were neither representative nor conclusive, as can be seen in the following graph, which we include for reviewer 1. We therefore searched for alternative, more sensitive techniques, such as dPCR.

Comparative mRNA relative expression between organoids at three stages (early, intermediate, and late) for selective epithelial populations: EPCAM (mature enterocytes), proliferative cells (PCNA), Stem cells (SOX9), and microvilli (Vil-1). Results were normalized to the tissue value (=1).

3. Regarding the dPCR results, no specific statistical analysis is provided. The authors discuss variability across mice but it is not clear if the variability derives from biological differences or if it is intrinsic to the method used. To compare this, technical replicates are required on a per-sample basis. If the material is not sufficient, one should at least compare organoids from the same animal and not different animals or time-points. Without this, the comparisons discussed in the paper across animals and organoid stage are not statistically solid. Also, GADPH was detected as housekeeping gene but not used for any normalization in the results.

Our study is focused on exploratory research to identify tissue-specific transcripts and record trends. The lack of technical replications is due to the limited biological material, and this is also the reason why we do not perform an exhaustive statistical analysis. Organoids from the same animal were compared in the different stages, but results could not be transferred to other animals. However, the results provide valuable information recording the expression of tissue-specific transcripts even with very low concentrations of RNA. GAPDH was used as a qualitative marker of the integrity of the samples; therefore, no normalization of genetic expression or determination of double change were done. Due to the biological variability between the samples inherent to their growth, it was not possible to normalize expression with this endogenous gene.

4. Several points can be addressed on the figures:

• Fig.1 was probably loaded incorrectly since it is the same as Fig.3. Therefore no comment can be added on Fig. 1, which nevertheless seems to be the main finding of the whole manuscript;

We thank the reviewer for pointing out this mistake. We now uploaded the correct Fig. 1 Absolute quantification of cell markers identified in duodenal organoids by dPCR. Comments to the figure are included from line 212-256.

• (b) Fig.2 shows different epithelial markers as the one included in the figure legend. Also, it does not include any “No template control” as introduced in the figure legend.

We now describe the correct markers in the figure legend, adding GAPDH, and also the scatterplots for TISSUE and NTC, as another reviewer suggested (lines 296-303).

• (c) In Fig.3 no scale bar is present and it is not discussed what the different arrows should point to.

The scale bar has been added, as is the description of arrows pointing to mitochondria, while arrowheads show tight junctions. Changes can be seen in lines 316-322.

• (d) Scale bars are also missing in Fig.4. (e) The authors should consider discussing Fig.3 and Fig.4 in more details in order to justify their appearance as main figures and therefore as main result of the manuscript. In the present state of the manuscript, no quantification is provided of the represented markers and the novelty of the results is not completely clear. If the figures only represent a confirmation of the expression of the markers, then it may be more fitting to include them as supplementary figures.

The scale bar is now added to Fig. 4.

Figures 3 and 4 in the manuscript are crucial to confirm the results obtained by dPCR, which is the innovation introduced in our manuscript. Both types of microscopies help to identify key elements at the ultrastructural and protein levels, confirming that the tissue-specific transcripts amplified by dPCR can be related to proteins visualized by confocal microscopy, and to the ultrastructural characteristics of the tissue in the organoids.

Minor points

1. In the Supplementary material 2 not all information about the used antibodies are listed.

We have now completed the information.

Reviewer #2:

1. Title:

It could be beneficial to slightly reword the title for clarity and engagement, such as:

- "Identification of Epithelial Markers in Murine Duodenal Organoids Using Digital PCR Technology."

- "Digital PCR Characterizes Epithelial Cell Populations in Murine Duodenal Organoids.

Thank you for the suggestions. We will use "Digital PCR Characterizes Epithelial Cell Populations in Murine Duodenal Organoids”. The new title is now in lines 1-2.

2. Materials and Methods

- A few areas could benefit from more specific information, such as the concentrations of growth factors used in the culture medium, which are crucial for reproducibility

Initially, this was described in supplementary material 1, but we have now moved this information to the Materials and Methods section 2.2 Organoid culture and maintenance; changes appear in lines 118-130.

- Several abbreviations are used without first providing the full terms. For instance, abbreviations such as DMEM (Dulbecco's Modified Eagle Medium), PBS (Phosphate Buffered Saline), and others should be introduced with their full names upon first mention. I recommend revising the text to include the full names of all abbreviations when they are first introduced.

We have added the full terms throughout the manuscript.

- Antibody incubation steps are clear, but the concentrations or dilutions of both primary and secondary antibodies should be included to ensure reproducibility. The concentration of Hoechst stain used in the experiments should be specified.

We specified the dilutions of primary and secondary antibodies in lines 196-197 and 202, and Hoechst dilution in line 199 of the manuscript. In addition, dilutions of primary and secondary antibodies are included in the S2 Supporting information, along with the description of the antibodies used.

- Add Specific Concentrations/Volumes: In several places, concentrations (especially for growth factors and antibodies) are missing. Including these values would make the protocol more precise.

The information was completed in Material and Methods and Supporting Information S2.

3. Supplementary material 2

- The size of the PCR product should be specified in base pairs (bp) for clarity and precision.

We added the missing information to Supporting information S2.

- The oligonucleotide sequences of the primers must be indicated in the 5′ to 3′ orientation to ensure correct interpretation and reproducibility.

We have now indicated the 5’ or 3’ orientation in Supporting information S2.

Reviewer #3:

There are several issues that need revision.

• In the methods part, it is not clear at all how the authors proceeded to grow organoids once they had isolated and cultured duodenal crypts. It is not enough to refer to other publications without making clear what procedure the authors used - at least this was unclear for me. Please make this unambigously clear in the manuscript.

We have detailed the procedure in section Material and Methods 2.2 Organoid culture and maintenance, in lines 118-130. Because Reviewer No. 2 also raised the same question, the text is marked with blue.

• 2. I definitely recommend to show the results obtained for GAPDH in the manuscript itself and NOT ONLY in the supplemental material. The extent of GAPDH expression is a crucial control and definitely needs to be shown so that readers can immediately compare the results for GAPDH with those obtained for cell specifically expressed genes.

Attending the reviewer´s suggestion, several changes were introduced in Fig. 2, including GAPDH expression, and tissue and NTC results of 3 examples. Complete scatterplots are included in Supporting Figure S1.

• 3. Unclear is, to what extent the organoids contained regular cells and to what extent they showed irregularities. Here the authors have to show what percentage of cells was regular and what percentage irregular, or how many structures (percentage-wise) were seen compared to original tissue. Also, here I would recommend to state whether these percentage values are in the range of the results that other authors have observed or not. A reader who is not that familiar with how organoids should look like would definitely need this information.

We understand the situation; however, the focus of this study is not to calculate the number of cells or organoids, but rather to see if expression of tissue-specific transcripts can be detected from samples with very low concentrations of genetic material and correlate this expression with the ultrastructure and the presence of key proteins in tissue and organoids. For this reason, we did not quantify structures or percentages compared with the tissues of origin.

• 4. In the last part of the manuscript (results/discussion), the authors name ddPCR, dPCR and QIAcuity dPCR. They do not explain what the main difference is. Though, it is crucial - otherwise the readers do not understand why the authors are claiming that their data and results are significant and worth to be published and made know to the scientific community. The differences between ddPCR, dPCR, QIAcuity dPCR need to be clarified with a few sentences so that also readers who do not use these methods can clearly understand the main differences.

We appreciate the suggestion. We now include some paragraphs comparing dPCR with ddPCR, you can find the new information in lines 410-424.

• 5. The legends needs to be more informative. What is shown exactly? in the micrographs it should be stated in the legend to what structures the arrows point to and why that is important.

Thank you for the suggestions. We made changes in figure legends, which you can find in lines 258-262 for Fig.1, in lines 296-303 for figure 2, in lines 316-322 for Fig. 3 and in lines 335-339 for figure 4.

---

## [Decision Letter · Decision Letter 1]

17 Dec 2024

PONE-D-24-39881R1Digital PCR Characterizes Epithelial Cell Populations in Murine Duodenal OrganoidsPLOS ONE

Dear Drs. Espinosa-Castellano,

Thank you for submitting your manuscript to PLOS ONE. After careful consideration, we feel that it has merit but does not fully meet PLOS ONE’s publication criteria as it currently stands. Therefore, we invite you to submit a revised version of the manuscript that addresses the points raised during the review process.

We look forward to receiving your revised manuscript.

Kind regards,

Alvaro Galli

Academic Editor

PLOS ONE

Journal Requirements:

Reviewers' comments:

Reviewer's Responses to Questions

**Comments to the Author**

1. If the authors have adequately addressed your comments raised in a previous round of review and you feel that this manuscript is now acceptable for publication, you may indicate that here to bypass the “Comments to the Author” section, enter your conflict of interest statement in the “Confidential to Editor” section, and submit your "Accept" recommendation.

Reviewer #1: All comments have been addressed

Reviewer #2: All comments have been addressed

2. Is the manuscript technically sound, and do the data support the conclusions?

Reviewer #1: Yes

Reviewer #2: Yes

3. Has the statistical analysis been performed appropriately and rigorously? 

Reviewer #1: No

Reviewer #2: N/A

4. Have the authors made all data underlying the findings in their manuscript fully available?

Reviewer #1: Yes

Reviewer #2: Yes

5. Is the manuscript presented in an intelligible fashion and written in standard English?

Reviewer #1: Yes

Reviewer #2: Yes

6. Review Comments to the Author

Reviewer #1: The aim of the manuscript is to identify key cellular components of murine duodenal tissue in organoid preparations using digital PCR (dPCR) technique. The results include dPCR results as well as electron microscopic and confocal characterization of duodenal organoids collected at different stages and accordingly classified into early, intermediate and late organoids.

The authors adressed the points raised in the previous round of revision and accordingly changed figures and text in an appropriate manner. The authors discuss that the lack of appropriate statistical analysis and technical replicates derives from the limited biological material. This may be true but still does not justify the description of comparisons between organoid stages as well as the use of terms such as "absolute quantification" in the text. The change of title suggested by another reviewer is in line with this consideration. We think that the use of dPCR as a qualitative exploratory research tool should be more clearly stated throughout the text.

Also, according to the authors, Fig. 3 and Fig. 4 should be a confirmation of the results obtained by dPCR. This was quite clear for Fig. 4 but it is still less obvious for Fig. 3. Comparisons between different cell populations and specific cellular structures should be more carefully described to justify this figure. At the time being, the figure seems to be inserted to prove the quality of the organoids produced, which is not the aim of the work presented in the manuscript.

Reviewer #2: (No Response)

7. PLOS authors have the option to publish the peer review history of their article (what does this mean? ). If published, this will include your full peer review and any attached files.

**Do you want your identity to be public for this peer review?** For information about this choice, including consent withdrawal, please see our Privacy Policy .

Reviewer #1: No

Reviewer #2: **Yes: ** Huda Hisham Sultan Alkatib

---

## [Author Response · Author response to Decision Letter 1]

24 Jan 2025

Reviewer #1: The aim of the manuscript is to identify key cellular components of murine duodenal tissue in organoid preparations using digital PCR (dPCR) technique. The results include dPCR results as well as electron microscopic and confocal characterization of duodenal organoids collected at different stages and accordingly classified into early, intermediate and late organoids.

The authors addressed the points raised in the previous round of revision and accordingly changed figures and text in an appropriate manner. The authors discuss that the lack of appropriate statistical analysis and technical replicates derives from the limited biological material. This may be true but still does not justify the description of comparisons between organoid stages as well as the use of terms such as "absolute quantification" in the text. The change of title suggested by another reviewer is in line with this consideration. We think that the use of dPCR as a qualitative exploratory research tool should be more clearly stated throughout the text.

Answer: The reviewer correctly mentions that we did not perform an exhaustive statistical analysis because of the lack of technical replications and the limited biological material. To address his concerns, we eliminated the phrase “absolute quantification” to avoid misunderstandings. In addition, we made some changes that can be seen the revised manuscript with track changes in lines 66-71, 224-229, 238-242, 280-281, 408-409,433-434.

Also, according to the authors, Fig. 3 and Fig. 4 should be a confirmation of the results obtained by dPCR. This was quite clear for Fig. 4 but it is still less obvious for Fig. 3. Comparisons between different cell populations and specific cellular structures should be more carefully described to justify this figure. At the time being, the figure seems to be inserted to prove the quality of the organoids produced, which is not the aim of the work presented in the manuscript.

Answer: Transmission electron microscopy (TEM) plays a crucial role in understanding organoid development because it provides highly detailed, high-resolution images of cellular structures. In our experience, TEM allows us to observe the structural integrity of the different cell populations within organoids as they develop, at better resolution than conventional light microscopy. Additionally, transmission electron microscopy makes it possible to track insights into the tissue architecture, and the structural relationship between different cell types. With the digital PCR characterization of epithelial cell populations in murine duodenal organoids that we are presenting in this article, we must provide evidence to the reader that the cells analyzed are not only functionally but also structurally healthy, through the expression of biomarkers measured by dPCR and confirmed by confocal microscopy for the former, and through TEM for the latter.

Reviewer #2: (No Response)

---

## [Decision Letter · Decision Letter 2]

6 Feb 2025

Digital PCR Characterizes Epithelial Cell Populations in Murine Duodenal Organoids

PONE-D-24-39881R2

Dear Drs. Espinosa-Cantellano,

We’re pleased to inform you that your manuscript has been judged scientifically suitable for publication and will be formally accepted for publication once it meets all outstanding technical requirements.

Kind regards,

Alvaro Galli

Academic Editor

PLOS ONE

Additional Editor Comments (optional):

Reviewers' comments:

Reviewer's Responses to Questions

**Comments to the Author**

1. If the authors have adequately addressed your comments raised in a previous round of review and you feel that this manuscript is now acceptable for publication, you may indicate that here to bypass the “Comments to the Author” section, enter your conflict of interest statement in the “Confidential to Editor” section, and submit your "Accept" recommendation.

Reviewer #1: All comments have been addressed

Reviewer #2: All comments have been addressed

2. Is the manuscript technically sound, and do the data support the conclusions?

Reviewer #1: Yes

Reviewer #2: Yes

3. Has the statistical analysis been performed appropriately and rigorously? 

Reviewer #1: N/A

Reviewer #2: N/A

4. Have the authors made all data underlying the findings in their manuscript fully available?

Reviewer #1: Yes

Reviewer #2: Yes

5. Is the manuscript presented in an intelligible fashion and written in standard English?

Reviewer #1: Yes

Reviewer #2: Yes

6. Review Comments to the Author

Reviewer #1: (No Response)

Reviewer #2: Thank you for your careful revisions and for addressing all my previous comments. I have reviewed the revised manuscript, and I find that the authors have adequately incorporated the suggested changes. The manuscript is now well-structured, clear, and aligns with the journal's standards. I have no further major concerns.

7. PLOS authors have the option to publish the peer review history of their article (what does this mean? ). If published, this will include your full peer review and any attached files.

**Do you want your identity to be public for this peer review?** For information about this choice, including consent withdrawal, please see our Privacy Policy .

Reviewer #1: **Yes: ** Davide Gobbo

Reviewer #2: **Yes: ** Huda Hisham Sultan Alkatib

---

## [Editor Report · Acceptance letter]

PONE-D-24-39881R2

PLOS ONE

Dear Dr. Espinosa-Cantellano,

I'm pleased to inform you that your manuscript has been deemed suitable for publication in PLOS ONE. Congratulations! Your manuscript is now being handed over to our production team.

Kind regards,

on behalf of

Dr. Alvaro Galli

Academic Editor

PLOS ONE